# Characterization of some fungal pathogens causing anthracnose disease on yam in Cross River State, Nigeria

**Nkese Ime Okon[1], Aniedi-Abasi Akpan Markson[2], Ekeng Ita Okon[2], Effiom Eyo Ita[1], Edak Aniedi Uyoh[1], Ene-Obong Effiom Ene-Obong[1], Valentine Otang Ntui[1,3]***

**1** Department of Genetics and Biotechnology, University of Calabar, Calabar, Nigeria, **2** Department of Plant and Ecological Studies, University of Calabar, Calabar, Nigeria, **3** International Institute of Tropical Agriculture, Nairobi, Kenya

* v.ntui@cgiar.org, ntuival@yahoo.com

**Data Availability Statement:** All relevant data are within the paper and its Supporting Information files.

## Abstract

Yam anthracnose is one of the most serious fungal diseases affecting white and water yam production. Screening of available landraces for new sources of durable resistance to the pathogen is a continuous process. In the present study, the pathogens causing anthracnose in *Dioscorea alata* and *Dioscorea rotundata* farms in Cross River State yam belt region were characterized. Diseased yam leaves with anthracnose symptoms collected from the farms were used in the isolation, purification and, identification of *C. alatae* strains using morphological, cultural, and molecular methods. Leaf chlorosis, leaf edge necrosis, blights, dark brown to black leaf spots, shot holes, necrotic vein banding and vein browning were the predominantly observed symptoms. Seven isolates of *C. alatae*, Ca5, Ca14, Ca16, Ca22, Ca24, Ca32 and Ca34, and one isolate of *Lasidioplodia theobromae*, Lt1 were found to be associated with yam infection in Cross River State, with Lt1 as the most prevalent, occurring in all the locations. These isolates were classified into four forms which included the slow-growing grey (SGG), the fast-growing grey (FGG), the fast-growing salmon (FGS), and the fast-growing olive (FGO). Sequence analysis of the ITS region revealed <80% nucleotide identity between the isolates and the reference C. *gloeosporioides*. Pathogenicity test showed that all the isolates displayed typical symptoms of anthracnose disease as were observed in the field, but Lt1 was the most virulent. Inoculation of 20 *D. alata* and 13 *D. rotundata* landraces with isolate Lt1, showed that 63.64% of the landraces were susceptible while 36.36%were resistant. *D. alata* landraces were the most susceptible. This study revealed that anthracnose is prevalent and may assume an epidemic dimension in the yam growing communities of the state. There is need for increased effort in the breeding of yam for anthracnose resistance.

## Introduction

Yams (*Dioscorea* spp.) are monocotyledonous plants with underground tubers and constitute the predominant starchy staple in sub-Saharan Africa especially in the five West African

**Funding:** This research was supported by the Nigerian Tertiary Education Trust Fund (TETFund) grant TETFUND/DR&D/CE/NRF/2016/STI/13/VOL.1 awarded to the Department of Genetics and Biotechnology, University of Calabar, Calabar, Nigeria, for "Production of anthracnose-resistant yam seedlings for use by Nigerian farmers. The funders had no role in study design, data collection and analysis, decision to publish, or preparation of the manuscript.

**Competing interests:** The authors declare that no competing interest exist.

countries (Nigeria, Cote d'Ivoire, Ghana, Benin, Togo) widely regarded as the 'Yam belt region' [1, 2]. There are about 600 known species of yam, of these, only 11 are edible, and six out of these are cultivated and consumed in Nigeria [2, 3]. Among these six species, white yam (*Dioscorea rotundata*) commands the highest market value owing to the superiority and preference of its tubers for food in Nigeria [4].

Nigeria has been consistent in the lead of yam producing countries worldwide. Available statistics indicate that in 2012, Nigeria accounted for over 65% (38 million tons) of the over 58.8 million tons of yams produced, valued at $7.75 billion and cultivated on 2.9 million hectares [5]. Recently, Nigeria's annual yam production statistics stands at over 45,004 million metric tons [6]. Despite the increasing yam production figures reported for Nigeria [5–7], yam production is generally constrained by a myriad of problems some of which are weed pressure, decline in soil fertility, storage pests, high cost of labour, cost of land preparation and maintenance, staking and barn making and, most importantly, field pests and diseases. Among the diseases plaguing yam, anthracnose or die back disease is one of the most ravaging [2, 8], limiting white yam (*D. rotundata*) and water yam (*D. alata*) production in the tropics [2, 9]. In Nigeria, *Colletotrichum* disease complex (commonly referred to as anthracnose or die-back) remains one of the most challenging and destructive diseases, causing heavy losses in yam [10]. Although *Colletotrichum* species have been implicated as the key pathogens of dieback, *Lasiodiplodia theobromae* is also reported to cause dieback of many crops including yam, resulting in heavy losses to farmers [10, 11]. Hence, it is seen as an important pathogen component of the anthracnose disease complex. Anthracnose has been implicated in yam tuber yield losses ranging from 50 to 90% under favorable conditions for pathogen infection, establishment, and disease development [8]. Infection and disease symptoms are concentrated on leaves, though yam petioles, stems, and tubers are also known to be infected. The capability of *Colletotrichum* and *L. theobromae* for multiple routes of transmission and their ability to persist in the field are the major strengths of these pathogens for extensive crop damage. The pathogens can be transmitted from foliage to tuber, and from tuber to foliage in the following season [10] while overwintering in infected planting material (stored tubers). Alternative hosts and crop debris constitute the main sources of pathogen inoculum [12, 13].

The use of fungicides and traditional control methods adopted by farmers are transient in their effects. Efforts are ongoing in Nigeria, India, Ghana, Ivory Coast, Guadeloupe and Vanuatu towards obtaining anthracnose resistant hybrids [2]. However, the use of genetic engineering tools (such as genome editing) to complement conventional breeding techniques is advocated [2].

Reports on extensive survey of anthracnose disease incidence and severity on yam caused by *Colletotrichum* species in Nigeria are scanty especially, in the South-South geopolitical zone of the country. However, a 61.7% occurrence of *Colletotrichum* species among seven other targeted pathogens and an 84.2% occurrence of *Colletotrichum* spp. among species of *Colletotrichum* isolated were reported in the southern guinea savanna and the southern forest area of Nigeria [9]. Symptoms vary based on yam species and ecological region of occurrence [10]. Typical symptoms exhibited by most *D. alata* cultivars grown in the Southern Guinea savanna region are circular black spots on leaf surfaces, expanding to express leaf edge necrosis and then progressing to vine blackening and tip die-back. On the other hand, typical symptoms reported for *D. rotundata*, are black circular spots randomly distributed on leaf surfaces with extensive defoliation and vine blackening from severe infection. In the southern forest region, symptoms on *D. alata* were said to appear mostly as streak browning lesions, starting off on leaf veins and rapidly expanding to cover the entire leaf [10]. Comparatively, impact of anthracnose disease on yam is often rated higher than that of viruses. A 72% disease severity was reported for anthracnose in six genotypes of *D. alata* in Ibadan, Western Nigeria over that

of some viruses under similar test conditions [14]. Also, an extensive assessment of field occurrence of *C. gleosporioides* in Benue state has been reported [15].

Cross River is one of the major yam-producing states in Nigeria. The objective of this study was to characterize and identify *Colletotrichum* isolates, and other key pathogen(s) associated with yam anthracnose in Cross River State, Nigeria and to determine the relationship among them as well as their virulence. Results obtained should help proffer solutions on how to manage the disease.

## Materials and methods

### Study area

The study area constitutes the three senatorial districts of Cross River state covering the southern guinea savanna region to the north and the humid rain forest agro-ecological zone to the south. The northern part of the state where yam-producing communities like Ogoja, Yala and Bekwara are located lies within the southern guinea savanna region while the southern part of the state (Obubra, Yakurr, Ikom, Akpabuyo and Calabar South) is in the humid rainforest agro-ecological zone. Cross River has an annual temperature fluctuating between 22˚C and 32˚C and annual rainfall of over 2000 mm and is located within longitude 4˚ 57' 0" North and latitude 8˚ 19' 0" East. Cross River State is agrarian with a greater population of her inhabitants predominantly engaging in farming, particularly, yam, cassava, banana, and plantain cultivation.

### Collection of infected yam leaves from sampling locations

The sample area was surveyed using cluster sampling method. A total of 21 locations were sampled in six Local Government Areas across the three senatorial districts of Cross River State. The Local Government Areas were the clusters where farms were identified and sampled in selected villages (units). Villages engaged in commercial production of yam were selected for the survey using simple random sampling procedures allowing for equal chances of being selected from each cluster. In each village about 5 to 6 farms were visited. The farms were surveyed for *D. alata* and *D. rotundata leaves* with die back and necrotic lesions symptoms. Infected leaf samples were collected, labeled, and packaged accordingly for subsequent laboratory studies. Sampled areas and farm locations are presented in Table 1.

### Collection of yam landraces from sampling locations

Yam Landraces of *D. alata*. L and *D. rotundata* Poir were collected in September 2019 from local farmers in Cross River State and Benue State. Yam accessions were also obtained from National Root Crop Research Institute (NRCRI), Umudike, Abia State, Nigeria. A total of 20 *D. alata* and 13 *D. rotundata* landraces/accessions were collected (Table 2). Each yam tuber was packaged separately in a plastic bag and labelled accordingly. The tubers were stored in a well-ventilated room to break dormancy. Once sprouting was noticed, the tubers were cut into setts of average weight 350 g [16]. The cut setts were treated with wood ash before planting. Each of the yam landraces and accessions were replicated thrice.

### Isolation and identification of isolates from infected water yam leaves using cultural and morphological characteristics

The symptomatic yam leaves were surface sterilized with 0.1% sodium hypochlorite for three minutes and 70% alcohol for one minute. The leaves were rinsed three times in changes of sterile distilled water. Four pieces (5mm diameter) of each infected leaf tissues were cut with

**Table 1. Locations where yam farms were surveyed, and symptomatic leaves collected.**

| S/N | Sampling Areas | Farm locations |
| --- | --- | --- |
| 1. | Obubra | Ovonum |
| | | Ofodua |
| | | Ochon, |
| | | Crutech, Obubra campus |
| 2. | Yakurr | Ugep Town |
| | | Ntankpo |
| | | Convent villages |
| 3. | Ogoja | Ndok |
| | | Mbok |
| | | Ukpe |
| | | EgojaNdim |
| | | Ekajuk |
| | | Okundi |
| 4. | Ikom | Nde |
| | | Ikom town |
| | | Edor |
| | | Okuni |
| | | Nkonfab |
| | | Alise |
| 5. | Akpabuyo | Akansoko |
| 6. | Calabar South | University of Calabar farms |
| | | New Airport farm |

flame-sterilized scalpel and inoculated on the solidified potato dextrose agar (PDA) medium in different plates. The inoculated plates were stored at room temperature (28°C) and observations were made daily for emergence of colonies. Sub-culturing was done weekly to obtain pure cultures of the isolates. The isolates were stored on slants of potato dextrose agar (PDA) in properly corked 250 ml conical flasks. Subcultures were made in petri dishes using PDA when the need arose [17]. Cultural and morphological characteristics by which the isolates were identified included mycelia colour, growth pattern, nature of mycelia and growth rate in the Petri dish. All these characters except growth rate were studied through visual appreciation in comparison with structures in standard reference atlas of imperfect fungi by Barnett and Hunter [17] and with literature on identification of *Colletotrichum* species by Abang et al. [18–20].

## Identification of isolates using molecular methods

**DNA extraction.** DNA extraction from the fungal mycelia was done using a modified CTAB protocol [21]. The mycelia were harvested by filtration through mesh sieves (40μL), washed with sterile distilled water, placed on Whatman filter paper to remove excess water and ground to a fine paste in 400 μL of extraction buffer using mortar and pestle. The ground tissue was then put in microfuge tubes and incubated in a water bath at 65°C for 15 min, followed by centrifugation at 12000 rpm for 5 min. Four hundred (400) μL of the supernatant was transferred into new Eppendorf tubes and 250 μL of chloroform: isoamyl alcohol (24:1, v/v) was added, mixed by inversion, and centrifuged at 13,000 rpm for 10 min. The upper aqueous phase was transferred into clean microcentrifuge tubes and 50 μL of 7.5 M ammonium acetate, followed by 400 μL of ice-cold ethanol were added to each tube to precipitate the

**Table 2. *Dioscorea alata* and *Dioscorea rotundata* yam Accession/landraces and collection locations.**

| Code | Local name | Status | Species | Location |
|---|---|---|---|---|
| TDa 1100193 | - | Accession | D. alata | NCRI |
| TDa 1100010 | - | Accession | D. alata | NCRI |
| TDa 07100154 | - | Accession | D. alata | NCRI |
| TDa 1100432 | - | Accession | D. alata | NCRI |
| CA5 | Mkpasipiba | Landrace | D. alata | Calabar |
| CA6 | Ebeghe | Landrace | D. alata | Akpabuyo |
| CA7 | Efut | Landrace | D. alata | Akpabuyo |
| CA8 | Oboneje | Landrace | D. alata | Obubra |
| CA9 | Olele | Landrace | D. alata | Obubra |
| CA10 | Akabriinyang | Landrace | D. alata | Akpabuyo |
| CA11 | Ekautat | Landrace | D. alata | Akpabuyo |
| CA12 | Ebeghudukikot | Landrace | D. alata | Akpabuyo |
| CA13 | Obana | Landrace | D. alata | Yala |
| CA14 | Obunaonhlor | Landrace | D. alata | Yala |
| CA15 | Obunaolieyi | Landrace | D. alata | Yala |
| CA16 | Ogi 1 | Landrace | D. alata | Ishiagu |
| NA17 | Okpolukata | Landrace | D. alata | Yala |
| CA18 | Obana 2 | Landrace | D. alata | Yala |
| CA19 | Obunaigele | Landrace | D. alata | Yala |
| CA20 | Obunaochokpa | Landrace | D. alata | Yala |
| CR1 | Ogboja | Landrace | D. rotundata | Obubra |
| ER2 | Agbaocha | Landrace | D. rotundata | Ishiagu |
| ER3 | Orume | Landrace | D. rotundata | Ishiagu |
| ER4 | Iguma | Landrace | D. rotundata | Ishiagu |
| ER5 | Nka | Landrace | D. rotundata | Ishiagu |
| BR6 | Tiv yam | Landrace | D. rotundata | Benue |
| ER7 | Obiauturugo | Landrace | D. rotundata | Ishiagu |
| CR8 | Obubra | Landrace | D. rotundata | Obubra |
| TDr 11100873 | - | Accession | D. rotundata | NCRI |
| TDr 1000006 | - | Accession | D. rotundata | NCRI |
| CR11 | Ajaba | Landrace | D. rotundata | Yala |
| CR12 | Fakita | Landrace | D. rotundata | Yala |
| CR13 | Ijibo | Landrace | D. rotundata | Yala |

DNA. This was then mixed by slow inverted movements that caused the DNA to precipitate at the bottom of the tubes. The tubes containing the DNA were centrifuged at 13,000 rpm for 5 min followed by decantation. The DNA was washed twice with 0.5 mL of 70% ethanol by centrifuging at 15,000 rpm for 5 min. The DNA was then dried under the laminar hood and 50 µL of TE buffer was added to dissolve it and stored at—20˚C until required.

**PCR amplification of ITS-rRNA genes.** Nucleotide sequences for Internal Transcribed Spacer (ITS)-ribosomal RNA (rRNA) genes of *C. gloeosporioides* were downloaded from NCBI and aligned to identify conserved regions. As there were no conserved regions, three pairs of primers flanking the ITS1-ITS2 regions were designed from three different strains of *C. gloeosporioides*. Primer 1 was designed from *C. gloeosporioides* strain E6 with accession number KT325567.1. Primer 2 was designed from *C. gloeosporioides* strain C16 with accession number KC010547.1. Primer 3 was designed from *C. gloeosporioides* strain D1 accession number KT325559.1. The primer sequences (S1 Table) were sent to Inqaba for synthesis.PCR was done

in a20 μL reaction volume made up of 14.5 μL distilled water, 2 μL of 10x PCR buffer,0.4 μL of dNTP mix, 0.5 μL each of 10 mM forward and reverse primer, 0.1 μL of Hotstar Taq polymerase and 2 μL of DNA. PCR amplifications were performed using the following conditions: initial denaturation at 95˚C for 30seconds followed by 35 cycles of denaturation at 94˚C for 30 seconds, annealing at 55˚C for 30 seconds, extension at 72˚C for 1 minute and a final extension at 72˚C for 7 minutes. A total of 8 isolates were used for PCR analysis.

**Sequencing of the ITS region.** The PCR products were purified with QIAquick PCR purification Kit (Qiagen) according to the instructions in the user manual. The purified products were used to make PCR sequencing reaction. The sequencing reaction mixture consisted of 2.5 μL PCR product, 1.5 μL of 5X sequencing buffer, 0.5 μL of Big Dye Terminator, 0.5 μL of 10mM of either YamCgITS2_F or YamCgITS2_R primer and 5μL of nuclease free water. PCR amplification was performed in a thermal cycler with initial denaturation step at 96˚C for 2 min followed by 40 cycles of denaturation at 96˚C for 10 s, annealing at 50 ˚C for 10 s, extension at 60˚C for 4 min, and final extension at 72˚C for 4 min. After PCR amplification, the sequencing product was purified by adding 50 μL of 100% ethanol, 2 μL of 3 M Sodium Acetate and 2 μL of 125 mM EDTA, and incubated at room temperature for 30 min. The mixture was centrifuged at 4000 rpm for 25 min at 20 ˚C. The pellets were washed with 70% ethanol and air dried for 20 min in the hood. The pellets were then resuspended in 10 μL HiDi formamide and incubated at 65 ˚C for 5 min, 95 ˚C for 2 min and cooled in ice. The samples were then sequenced using ABI 3130 DNA sequencer (Applied Biosystems, California, USA). The sequences were assembled, edited, and analyzed using SnapGene software (WWW.snapgene.com). For each isolate, 4 replicates each for forward and reverse primers were sequenced.

**Phylogenetic analyses.** The nucleotide sequences for each isolate were used in BLASTn searches against the GenBank database (http://www.ncbi.nlm.nih.gov/BLAST) to identify the most similar sequences available in the database. There were high variations in the nucleotide identities therefore sequences in the GenBank that showed the highest similarity (>60%) to the isolates were used for alignment. Sequences were aligned using the MUSCLE program and used subsequently for phylogenetic analyses based on the Maximum Likelihood method with 1000 bootstrapping. *C. gloeosporioides* isolate with accession number KC010547, from which the primers were designed was included in the analysis.

**Pathogenicity test (Koch's Postulate).** The pathogenicity of isolates from the infected water yam leaves was tested. Disease-free *D. alata* and *D. rotundata* yam tubers were planted in pots containing sterilized soils in the screen house maintained at normal ambient environmental temperature. Two months old leaves were dusted, rubbed gently with carborundum and sprayed with inoculums (1.0 x 10$^4$ spores/ml) using a 450 mL atomizer spray gun. The inoculum was prepared using a 10-day-old pure cultures of *C. alatae* which was prepared and stored. The Petri dish containing the pure culture of the pathogen was flooded with distilled water and a sterile blade was used to scrape the mycelia off the solid medium to release the conidia into a 200 mL sterile beaker. The mixture was vigorously shaken using a magnetic shaker before filtering it through sterile cheese cloth to obtain spore (conidia) suspension. Spore (conidia) load of 1 x10$^6$ per mL was prepared through serial dilutions and spore count was done using haemocytometer [22]. Each inoculated leaf was covered with transparent polythene bag for 24 hours. The set up was monitored every day for symptom expression.

## Screening of yam for tolerance to *Lasiodiplodia theobromae*: Whole plant bioassay

Based on the pathogenicity test result, isolate Lt1 which appeared to be *L. theobromae* was the most virulent, so we decided to screen the yam accessions against this isolate. The 33yam

landraces or accessions, three triplicates each, were screened in a screenhouse for resistance against Lt1. Whole plant inoculation was done according to the method of Kolade et al. [22] with some modifications. Two months after planting, five fully opened leaves were inoculated by dusting and gently rubbing with carborundum. The wounded region was rinsed by spraying with distilled water using a 450 mL plastic atomiser spray gun. The leaves were then dipped in the inoculum, prepared as above, for 5 minutes. Each inoculated leaf was covered with moistened polythene bags for 24 hrs to aid disease development [23]. Thereafter, the leaves were opened and observed for symptom development every alternate day, but data were recorded at 16 weeks post inoculation (wpi) [24]. The plants were maintained for 6 months in the screenhouse in polythene bags replicated thrice (3 plants per landrace/accession). The percentage of the leaf area exhibiting lesions was estimated for each accession or landrace and used for classification into various tolerance levels. This was done using a five-scale range where 0 = no infection, 1 = 1–20%, 2 = 21–40%, 3 = 41–60%, 4 = 61–80% and 5 = 81–100% infection. The experiment was laid out in Completely Randomized Design (CRD) and Disease incidence (DI%) per plant was computed using the formula:

$$DI\ (\%)\ per\ stand = \frac{No.of\ leaves\ infected\ X\ 100}{Total\ number\ of\ leaves}$$

## Ethical statement

This work did not require ethical approval.

## Results

### Symptoms and distribution of *anthracnose*

Anthracnose disease symptoms were observed in all sites surveyed indicating the disease is widely distributed across the three senatorial districts. However, the symptoms were more prevalent in the central region, particularly, Yakurr compared to the other regions. Assessment of infected water yam stands in the sampled farms revealed a wide range of symptoms (Fig 1). The symptoms varied from leaf tissue necrosis to shoots die-back, appearing as extensive scotch or blight, progressing from leaf margins towards the centre of the leaf lamina. Also, some symptoms appeared as yellow margins surrounding dark brown to black necrotic leaf tissues. Some expanding leaves were observed to be twisted, especially at the apex. Some water yam stands exhibited leaf chlorosis and stunted growth. In Obubra sampling locations, symptoms included chlorosis (a), dark brown spot dotting the leaf lamina (b), enlarged white spot encircled by brownish ring (c), brown necrotic tissues affecting the leaf base around the petiole attachment portion (d). The predominant symptoms (e-l) in Yakurr area include, necrotic vein banding (e), brownish spot rimmed by yellow ring coalescing to form enlarged necrotic portions (g), chlorosis and blight encroaching from the margin towards the centre of the leaf (h), vein browning and leaf edge necrosis (i), necrosis of tissues resulting in large shot holes (j), chlorosis with leaf browning (k), leaf tissue bleaching causing twisted leaf apex (l). In Ogoja, the prevailing symptoms were chlorosis and blight encroaching from the margin towards the centre of the leaf (m), large shot hole encircled by brown necrotic ring rimmed by a yellow ring (n), brownish ring surrounding light brown to white central portion (p), enlarged white necrotic ring (q), leaves dotted with shot holes resulting from falling off of necrotic central portions (r), small brownish spot coalescing to form large brown necrotic areas on the leaves (s). The predominant symptoms displayed on leaves sampled from Akpabuyo were extensive blight and chlorosis (t), brownish to black spot merging into larger necrotic patches (u). In Calabar South, infected leaves were observed to exhibit leaf edge necrosis (v) and extensive

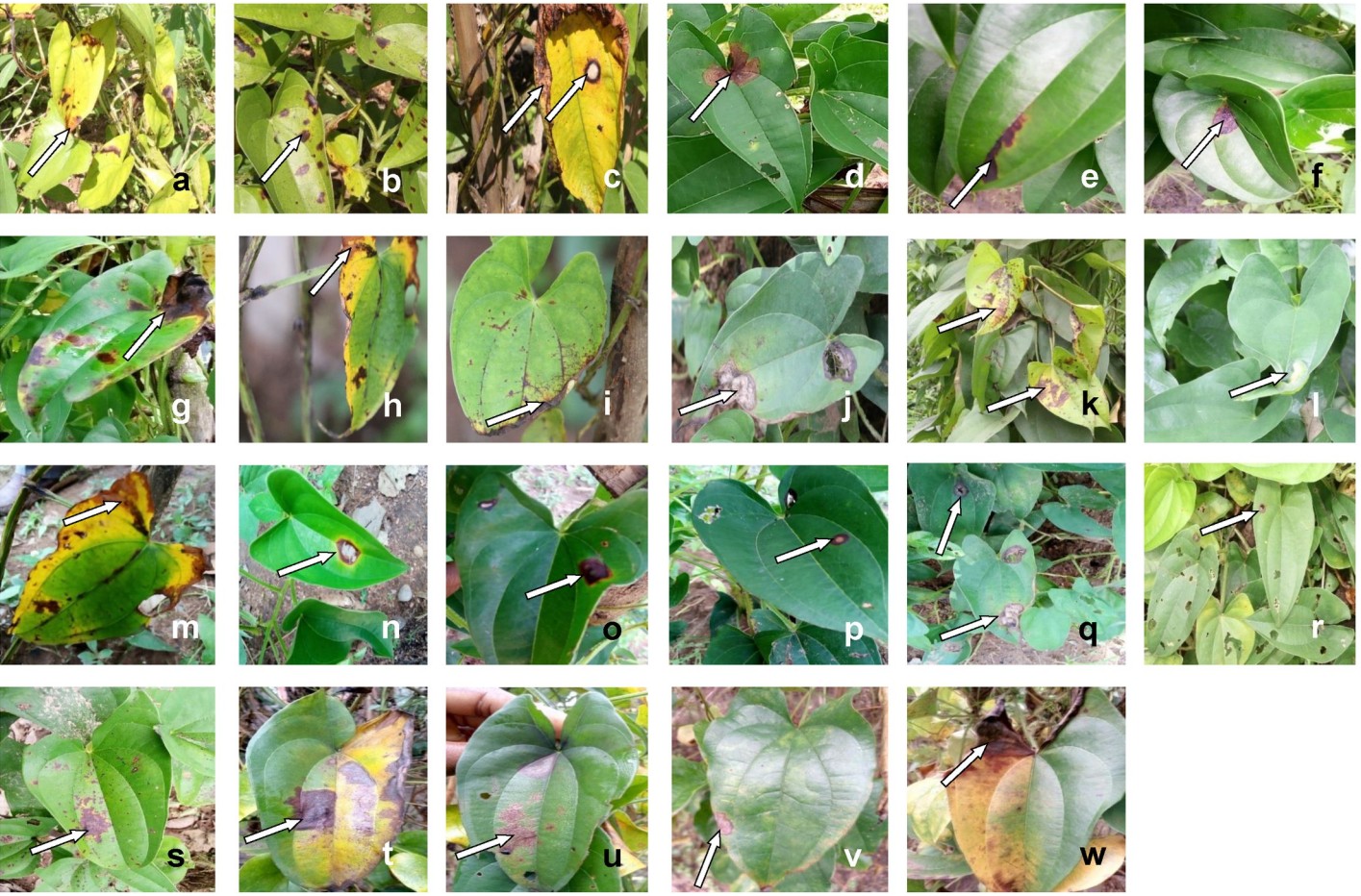

**Fig 1. Variation in appearance of anthracnose symptoms sampled on *D. alata* and *D. rotundata* in all the locations.** Infected leaves sampled from Obubra sampling area (a-d), Yakurr (e-l), Ogoja (k-q); Ikom (r-s), Akpabuyo (t-u), and Calabar South (v-x). Arrow indicates symptoms.

blight, chlorosis and tissue necrosis advancing from leaf margin towards the centre, often from one half of the leaf lamina.

## Identification of isolates from infected water yam leaves using cultural and morphological methods

Following isolations from the sample locations, seven isolates of *C. alatae*, Ca5, Ca14, Ca16, Ca22, Ca24, Ca32 and Ca34, and one isolate of *L. theobromae*, Lt1, were obtained (Fig 2) and described based on the cultural and morphological characteristics (Table 3). Observation of the pure cultures of the eight isolates, revealed striking culture characteristics of the mycelia in the growth medium. Ca5, Ca16, Ca24 and Ca34, had white mycelia which gradually turned grey with age while Ca14, Ca32 and Lt1had mycelial colours of orange, pink and cream, respectively. All except Ca16 exhibited radial and circular growth patterns with concentric rings while all the isolates displayed cottony mycelial growth except Ca16 and Ca32. Differences in mycelia growth pattern were also observed. This ranged from small to large concentric rings exhibiting mostly cottony growth mycelia (Table 3). The mycelial growth rate per day for the various isolates ranged from 1.69 to 3.92 mm and were within the growth rate range of 3.6 to 11.2 mm recorded for *C. gloeosporioides* [25, 26] and *L. theobromae* [11]. Based

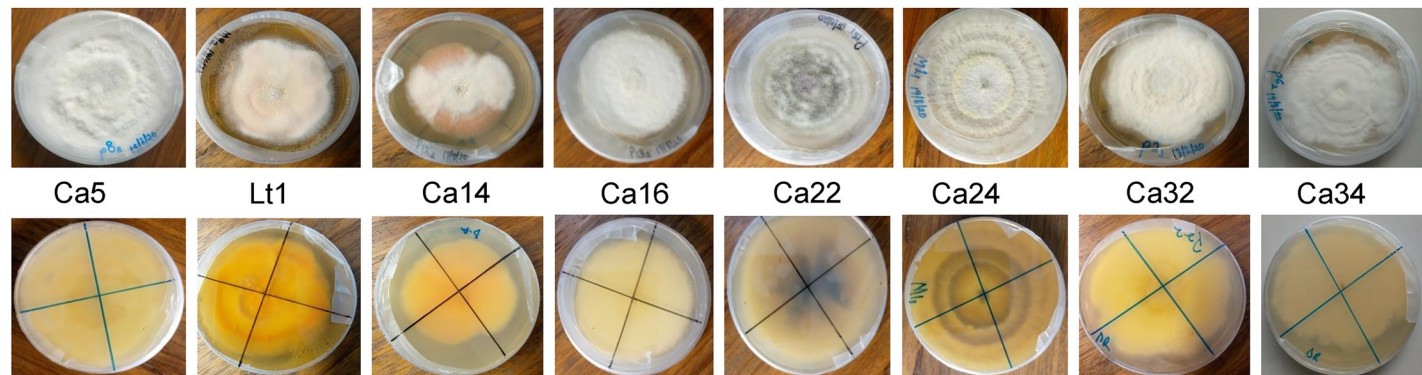

**Fig 2. Morphological variability of fungi isolates obtained from the different locations (top row, upper side of the colony; bottom row, reverse side of the colony).**

on the growth characteristics and literature, these isolates were confirmed to be *C. alatae and L. theobromae* and had four forms which included the slow-growing grey (SGG), the fast-growing grey (FGG), the fast-growing salmon (FGS), and the fast-growing olive (FGO) forms (Table 3). Cylindrical shaped conidia with rounded ends were recorded for some of the isolates (Fig 3). However, there were no distinct differences in the conidial appearance in terms of size and shape (Fig 3I). All were oblong or cylindrical, most were broadly rounded at both ends and some slightly tapering to the base. Other characteristics such as Acervuli were observed in the plates but not in infected specimens. These had dark spines (setae) at the edge of the structure and among the conidiophores (Fig 3J).

## PCR amplification

First, we selected 5 isolates and subjected them to PCR analysis using the three primer pairs shown in S1 Table. Only primer 2 (YamITS2_F and YamITS2_R) amplified sequences of the

**Table 3. Cultural characteristics of *Colletotrichum alatae* isolates.**

| *Colletotrichum alatae* isolates | Mycelia colour | Growth Pattern | Nature of mycelia | Colour on reverse side | Speed of growth | Mycelia growth rate |
|---|---|---|---|---|---|---|
| Ca5 | Whitish/Black | Large concentric ring | Cottony | yellow | Covered Petri dish in 7 days | 3.15 |
| Lt1 | Light pink to Orange | Small concentric ring | Cottony | Light pink | Covered less than 2/3 of Petri dish in 10 days | 1.69 |
| Ca14 | Dark pink | Circular growth | Cottony | Light pink | Covered less than 2/3 of Petri dish in 10 days | 1.84 |
| Ca16 | Whitish to Black / Fast-growing grey (FGG) | Plain | Wooly | Dark Grey | Covered Petri dish in 3 days | 3.92 |
| Ca22 | Brownish black | Large concentric ring | Cottony | Light grey | Covered Petri dish in 9 days | 2.32 |
| Ca24 | White/grey / Fast-growing grey (FGG) | Large concentric ring | Cottony | Grey | Covered Petri dish in 8 days | 2.85 |
| Ca32 | Creamy/light pink / Fast-growing salmond (FGS) | Small concentric ring | Velvety cottony | white | almost Covered Petri dish in 10 days | 2.74 |
| Ca34 | white | large concentric rings | Cottony | white | Covered 2/3 of Petri dish in 10 days | 2.1 |

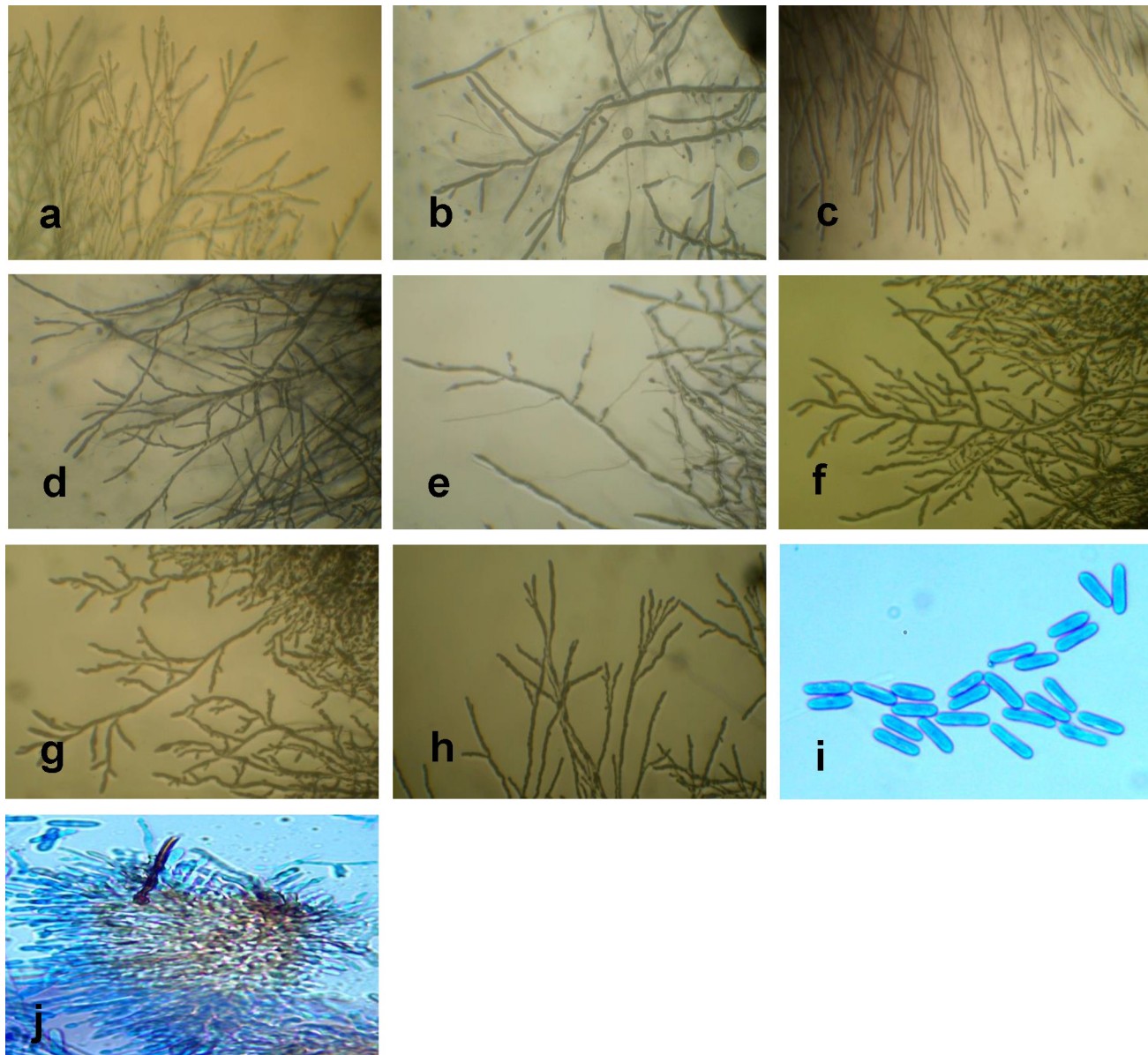

**Fig 3.** Photomicrographs showing growth pattern of fungi isolates from *D. alata* (a-e) and *D. rotundata* (f-h) at 10x.a(Ca5), one sided branching, non-branched tip, distance between branches ranges from 0.5–1.7cm, short bead-like hyphae with swollen tip. b (Lt1), alternate branching, non-branched tip, distance between branches ranges from 0.5–2.0cm, long cylindrical hyphae with tapering tip. c (Ca14), alternate branching, double branched tip, distance between branches ranging from 0.5–2.0cm, long branched thread-like hyphae with slightly swollen tip. d (Ca16), alternate branching with double branched tip, distance between branches ranges from 1.0–2.5cm, long branched thread-like hyphae with swollen tip. e (Ca22), alternate branching with non-branched tip, distance between branches ranging from 0.3–2.5cm, long branched cylindrical hyphae with tapering tip. f (Ca24), alternate branching, double branched at tip, distance between branches of about 0.5cm, short bead-like hyphae with slightly swollen tip. g (Ca32), alternating after two branches, triple branched at tip, distance ranging from 0.5–1.0, short bead-like branch with swollen tip. h (Ca34), alternating after two branches, multiple branched at tip, distance between branches ranging from 0.3–1.8cm, long bead-like hyphae with swollen tip. i, acervuli with dark spines (setae) at the edge of the structure and among the conidiophores. j, oblong and single-celled conidia.

ITS regions, including the 5·8S and 28S ribosomal RNA genes in all the isolates (Fig 4A). Primers 1 and 3 did not amplify isolate 11 (Lt1), therefore, we used primer 2 for amplification of the ITS1-ITS2 regions, in all the remaining isolates. PCR fragments of approximately 500 bp were

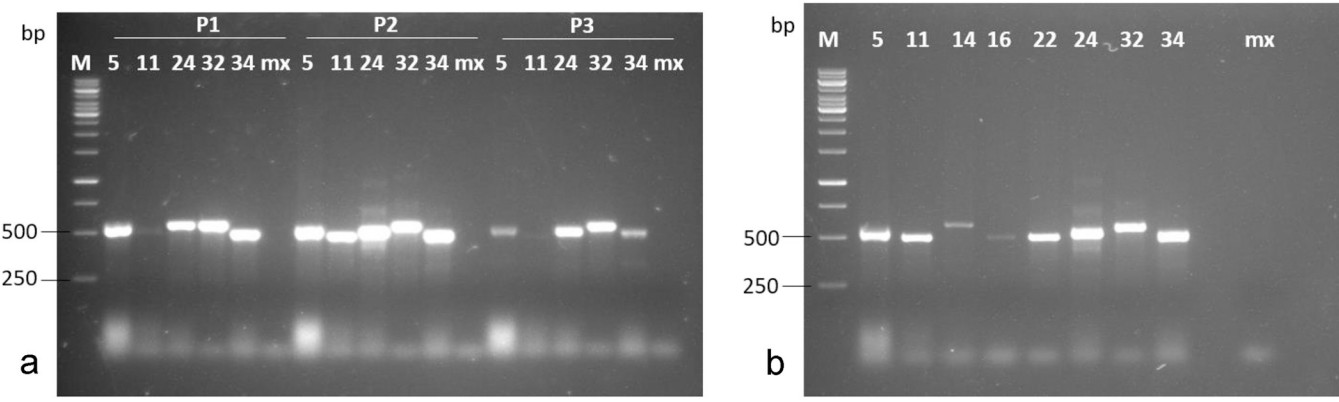

**Fig 4. PCR amplification of ITS gene from 8 isolates causing yam anthracnose.** a) PCR analysis using the three sets of primers. b) PCR analysis using primer set 2. 5,11,14, 16, 22 are isolates from *D. alata*. 24, 32 and 34 isolates *D. rotundata*. M, 1 kb DNA ladder, mx, DNA master mix.

obtained (Fig 4B). This result indicates that the isolated fungi are the same causative organism responsible for the manifestation of anthracnose symptoms in the leaf samples.

## Sequence analysis of the ITS region

PCR products obtained using primers YamITS2_F and YamITS2_R were purified and sequenced. The nucleotide sequences of the ITS region for six of the isolates were submitted to National Centre for Biotechnology Information (NCBI) database and were assigned the following accession numbers: OM365422 (Ca5), OM365423 (Lt1), OM365424 (Ca14), OM365425 (Ca24), OM365426 (Ca32) and OM427500 (Ca34). To determine whether the isolates are different, we checked the nucleotide identity between the isolates with the reference sequence using EMBOSS matcher-Pairwise Sequence Alignment (https://www.ebi.ac.uk/Tools/psa/emboss_matcher/). Alignment of the ITS regions revealed nucleotide sequence identities ranging from 60.1 to 76.2% between the isolates and the reference sequence (S2 Table). The isolates share <82% nucleotide identity (S2 Table) indicating that the isolates are equidistant and independent. The 5.8s rRNA was the most conserved genomic region across the isolates and the C. *gloeosporioides* reference genome with minimal variations, whereas the ITS1 and ITS2 were the most diverse.

Next, we compared the nucleotide sequences with those of published ITS sequences of several C. *gloeosporioides* and *L. theobromae* including the reference isolates. The evolutionary history was inferred using the UPGMA method. The percentage of replicate trees in which the associated taxa clustered together in the bootstrap test (1000 replicates) are shown next to the branches. The evolutionary distances were computed using the Maximum Composite Likelihood method and are in the units of the number of base substitutions per site. This analysis involved 29 nucleotide sequences. Codon positions included were 1st+2nd+3rd+Noncoding. All ambiguous positions were removed for each sequence pair (pairwise deletion option). Nucleotide sequences of *C. alatae* isolates were clearly distinguishable from C. *gloeosporioides* isolates (Fig 5). Isolates Ca5 (OM365422) and Ca14 (OM365424) were found in the same cluster with 81.4% nucleotide identity. Isolate Ca34 (OM427500) is an out-group with <70% identity with other *C. alatae* isolates but belong to the same cluster with C. *gloeosporioides* isolate KU097215. None of the *C. alatae* isolates was in the same cluster with the reference sequence KC010547.1 but were found clustering with other *C. gloeosporioides*. Isolate Lt1 (OM365423) was found clustering with *L. theobromae* isolates (Fig 5).

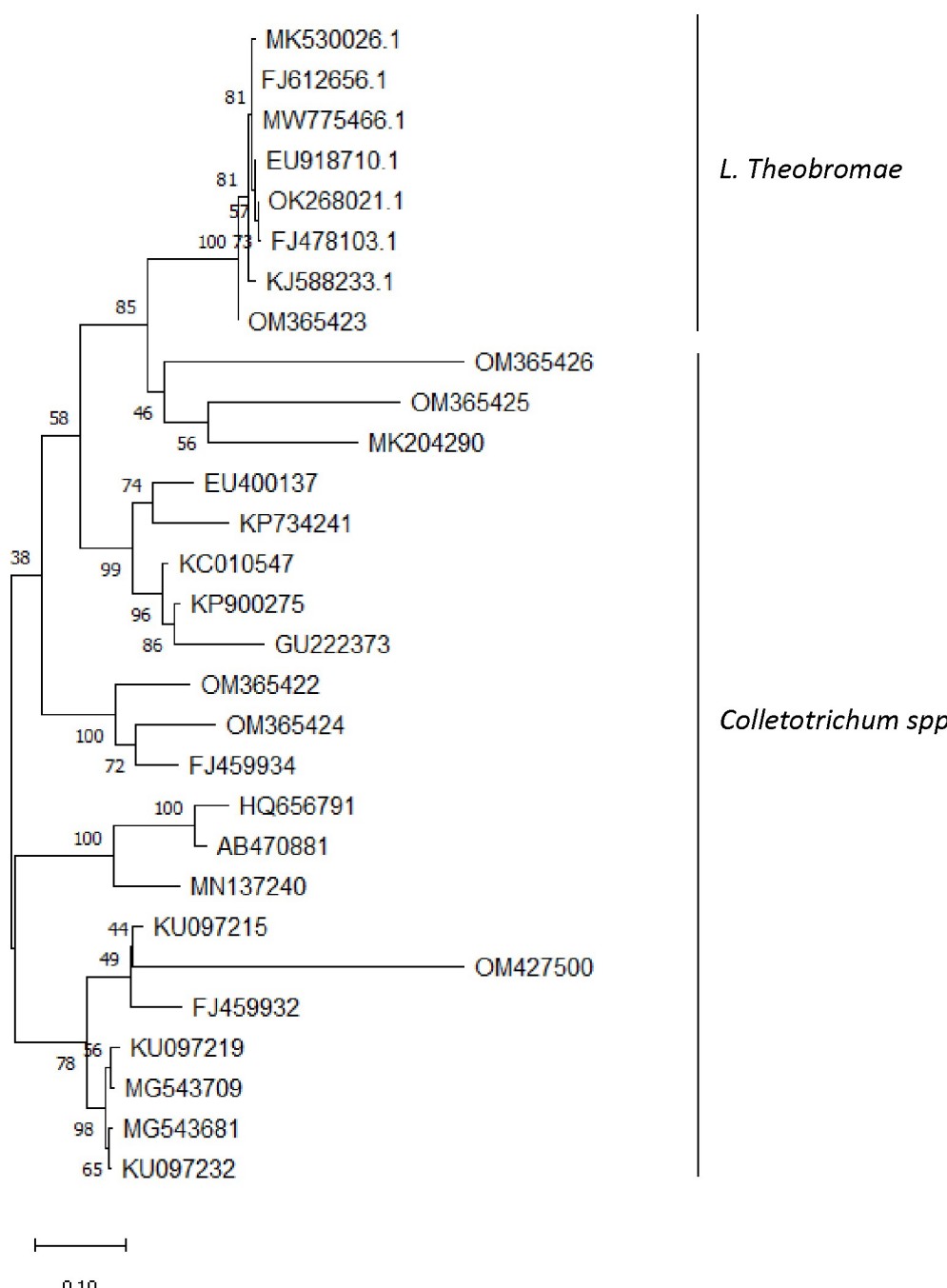

**Fig 5. Phylogenetic tree showing relationships of closely related accessions with our isolates using maximum likelihood method and based on the ITS gene sequences.** OM365422 (Ca5), OM365423 (Lt1), OM365424 (Ca14), OM365425 (Ca24), OM365426 (Ca32) and OM427500 (Ca34) are isolates obtained from this study.

## Pathogenicity test (Koch's Postulate)

The 8 isolates were subjected to pathogenicity test using susceptible water yam accessions. All the isolates displayed typical symptoms of anthracnose disease as were observed in the field. Anthracnose disease symptoms were not observed on the negative control leaves. However,

TDA1100193 was the most resistant, displaying insignificant level of susceptibility. On re-isolation, they all exhibited patterns of growth as observed in the original isolates. However, based on the prevalence of the pathogen in all sample locations, virulence of the pathogen and severity of infection during pathogenicity test, Lt1 was selected and used as the test pathogen to screen for resistance against all the collected landraces/accessions.

### Screening *D. alata* and *D. rotundata* for resistance to *Lasiodiplodia theobromae* in the screen house

All the twenty *D. alata* yam landraces and accessions screened for resistance against the isolate Lt1 exhibited varying levels of severity of infection (Table 4) and the symptoms started appearing two months after inoculation. TDA1100193 with a disease severity score of 1 was the only accession that was highly resistant (HR) to infection by *L. theobromae* representing only 5%. TDa 1100010 and CA14 with disease severity scores of 2 were considered as resistant to the *L. theobromae*. On the other hand, TDA 1100154, TDa 1100432 CA5, CA6, CA11, CA13, CA15, CA16 and CA18 were found to be susceptible to the pathogen and the landraces CA7, CA9 and NA17 were highly susceptible to the anthracnose strain (Table 4). In the resistant and highly resistant landraces/accessions only 20% and 30% leaves were infected, respectively and symptoms were only visible 9 weeks after inoculation. In *D. rotundata*, disease incidence ranged from 9% in BR6 to 95% in ER7 (Table 4). Among the 13 Landraces and accessions, 5, ER3, ER5, ER6, TDr100006 and CR12 were found to be highly resistant to the *L. theobromae* strain used while CR1, ER2, ER4, and CR11 were resistant. ER7 was highly susceptible while CR8 and NR9 were susceptible. The disease severity score ranged from 1 to 5 (Table 4).

## Discussion

About 7 types of spots and 4 types of blights have generally been identified as symptoms of dieback disease on yam leaves [27]. These symptoms are said to be in the form of dark lesions which are usually surrounded with yellow halo. Some are reported to be seen as dark brown rings surrounding a light brownish necrotic portion. In the present study, the yam farms visited presented a myriad of symptoms including the ones reported above. Other symptoms of anthracnose commonly expressed in white yam and water yam include leaf necrosis and dieback of vines [28]. The type of symptoms in a particular area seems to be substantially influenced by the prevailing ecological indices of the area. Reports of streak browning lesions starting off on leaf veins and rapidly expanding to cover the entire leaf was documented for *D. alata* in the forest region of Nigeria [10]. In the present study, necrotic vein banding, vein browning and leaf edge necrosis were common in Yakurr and Obubra (forest region). However, in Akpabuyo and Calabar South (also, in the forest zone) chlorosis and blight affecting the entire leaf margins progressing inwards were commonly observed. The contrasting nature of symptoms observed on the yams in the southern region of Calabar and Akpabuyo with those in earlier reports may be due to varietal differences in the yams under study. In the Southern Guinea savannas, symptoms were reported to commence with circular black spots on leaf surfaces and expanding to manifest as leaf edge necrosis and then progressing to vine blackening and tip die-back. Similar symptoms were commonly encountered in this study in Ogoja and Ikom, which fall within the same region. In addition, commonly displayed symptoms on infected water yam leaves included small brownish spots coalescing to form large brown necrotic areas on the leaves as well as large shot holes encircled by brown necrotic rings rimmed by yellow rings.

Isolates of fungal pathogens causing dieback disease are highly variable, and manifest a range of colony colours, growth rates and morphology [18, 29–34]. Three of the isolates in the

**Table 4. Disease incidence, disease severity score and resistance status in *Dioscorea alata* and *Dioscorea rotundata* landraces screened for anthracnose resistance.**

| Yam landraces | Disease incidence (%) | Disease severity score | Rating* |
|---|---|---|---|
| *D. alata* | | | |
| TDa 1100193 | 18.66 | 1 | HR |
| TDa 1100010 | 29.87 | 2 | R |
| TDa 07100154 | 70.41 | 4 | S |
| TDa 1100432 | 69.25 | 4 | S |
| CA5 | 77.46 | 4 | S |
| CA6 | 69.25 | 4 | S |
| CA7 | 85.91 | 5 | HS |
| CA8 | 41.66 | 3 | MS |
| CA9 | 81.08 | 5 | HS |
| CA10 | 54.10 | 3 | MS |
| CA11 | 70.13 | 4 | S |
| CA12 | 72.42 | 4 | S |
| CA13 | 71.42 | 4 | S |
| CA14 | 21.48 | 2 | R |
| CA15 | 72.39 | 4 | S |
| CA16 | 63.10 | 4 | S |
| NA17 | 92.94 | 5 | HS |
| CA18 | 69.86 | 4 | S |
| CA19 | 60.33 | 3 | MS |
| CA20 | 56.69 | 3 | MS |
| *D. rotundata* | | | |
| CR1 | 27.92 | 2 | R |
| ER2 | 3.18 | 2 | R |
| ER3 | 14.47 | 1 | HR |
| ER4 | 35.20 | 2 | R |
| ER5 | 12.76 | 1 | HR |
| BR6 | 9.52 | 1 | HR |
| ER7 | 90.44 | 5 | HS |
| CR8 | 77.52 | 4 | S |
| TDr 11100873 | 78.32 | 4 | S |
| TDr 1000006 | 19.01 | 1 | HR |
| CR11 | 33.26 | 2 | R |
| CR12 | 13.36 | 1 | HR |
| CR13 | 48.54 | 3 | MS |

*Accessions or land races with a disease incidence of 0–20% were given a score of 1and considered as highly resistant (HR), those with a disease incidence of 21–40% were given a score of 2 and regarded as resistant (R), those with a disease incidence of 41–60% (3), and regarded as moderately susceptible (MS), those with a disease incidence of 61–80% (4), and regarded as susceptible (S), and those with a disease incidence of 81–100% (5), and regarded as highly susceptible (HS). In general, indices >40% are susceptible.

present study bear similar morphological characteristics and exhibited comparable growth rates to two morphological forms described by [18]. Isolates Ca16 and Ca24 were comparable with the Fast-growing grey (FGG) and Ca32 similar to Fast-growing salmond (FGS). Several researchers have reported varying cultural characteristics ranging from mycelia colour, growth pattern, growth rate, nature of mycelia, colour changes with progressive growth and colour in

media. Colour variation of isolates have been reported from normal white to light grey, grayish brown, grayish white, greenish grey, pinkish and pinkish brown [30]. All the isolates obtained from infected yam in this study share these characters. Isolates Ca5, Ca16, Ca24 and Ca34 had whitish mycelia which gradually turned grey with age. All, except Ca16 exhibited radial and circular growth patterns with concentric rings while all the isolates displayed cottony mycelial growth except Ca16 and CA32. Colours in the range of creamy, orange and pink were exhibited by Ca4, Ca14 and Ca32.

Anthracnose disease is said to be caused by *Colletotrichum* disease complex [35]. Though the first report of anthracnose on water yam in Nigeria in 1980 by Nwankiti and Okpala [36] was credited to *C. gloeosporioides*, it is however known today that the disease is most often caused by a complex of seemingly similar forms of *Colletotrichum* together with some other fungal pathogens including *L. theobromae* [37]. This creates a challenge of identifying specifically, the cause of a particular infection in time. The systematics of the species complex of the genus, *Colletotrichum* since it was first reported has over the years been evolving based on the taxonomic tool employed by various scientists to characterize this group of fungi. *Colletotrichum* is the only genus of the family, Glomerellaceae and consists of some saprobes and endophytes [38] but dominated by pathogens causing diseases of virtually all categories of plant forms from fruits and vegetables [39–41] through cereals (grasses), pulses to root and tuber crops [10, 20, 42–44] in both tropical and temperate [45] regions of the world. Owing to the multiplicity of forms and the host-specific nature of this group of organisms, there has been a lot of erroneous categorization, misidentification, and naming.

To edge over these challenges, some researchers have resorted to using various specific morphological (taxonomic) characters to classify species in the genus and have come up with what is referred to as 'accepted species' [46, 47]. Winch et al. [48] and Abang et al. [18] reported that *Colletotrichum* isolates from diseased yam leaves were morphologically and genetically distinct but used a wide species concept to lump all yam isolates together under the name *C. gloeosporioides*. Weir et al. [35] found that yam anthracnose isolates from Nigeria, along with those from Barbados, Guadeloupe, and India, belonged to the same clade and matched the Slow-growing grey (SGG) group described by Abang et al. [17], and were thus classified as *C. alatae*. In Danzhou City, Hainan Province, China, anthracnose-like lesions discovered on the leaves of *D. alata* cultivar Da56 were morphologically and genetically like the SGG group found in West African yam and were named *C. alatae* [49]. While some authors have used the name *C. alatae* as the causal agent of yam anthracnose [35, 43, 49, 50], others [29] used the name *C. gloeosporioides*. Since the isolates described in this study are similar to those reported by Weir et al. [35] and Lin et al. [49], we refer to our isolates as *C. Alatae* as described by Ntui et al. [2].

The identification of *C. alatae* based on cultural, and morphological characteristics alone is not satisfactory as it can be mixed up with other species within the genus, most especially *C. Acutatum* [51, 52]. Moreover, different species of *Colletotrichum* can infect the same host and the foliage infection of *C. Acutatum* and *C. gloeosporioides* are difficult to differentiate in terms of their symptoms and cultural morphology [53, 54]; hence the need for the introduction of the molecular techniques for proper identification of the pathogen isolated in the present study. Polymerase Chain Reaction (PCR) amplification of the ITS region of fungal isolates in the present study gave amplicon size range of >500 bp which is in the same range with the findings of other researchers although with slight variations [53, 55]. The variations indicate polymorphism in the isolated strains of *C. alatae* whose actual identities were further revealed through sequencing of the ITS region genes. Comparative analyses of the sequences of the ITS region in the 6 isolates showed that isolates Ca5 and Ca14 were the most closely related with 81.4% nucleotide identity. Comparison of the ITS data from this study with *C. gloeosporioides*

sequences published in the GenBank showed limited nucleotide identity (<75%) indicating high diversity. Phylogenetic analysis showed some *Colletotrichum* species clustering with some of our isolates indicating there could be a mix infection in the field.

Isolate Lt1 was the most virulent, but it was one of the slow growing types in the plates, indicating that rapid growth in the plate is not proportional to virulence. A blast of this isolate on NCBI shows <75% nucleotide identity with other *C. gloeosporioides*, and >97% nucleotide identity with many *L. theobromae* isolates. *L. theobromae* is reported to be cosmopolitan in nature and has been reported to cause dieback infections on cash crops such as cocoa and yams [11, 56]. This and the fact that the isolate clustered in the same clade with *L. theobromae* supported by high boostrap value (Fig 5), confirmed its identity. Therefore, we named it *L. theobromae* isolate Lt1 with the accession number OM365423.When yam landraces/accessions were challenged with this isolate, a high number of landraces were susceptible to the pathogen suggesting *L. theobromae* infects yam. More studies are required to evaluate the prevalence of *L. theobromae* infection of yam in Nigeria. Nigeria, being a leading yam producing country in West Africa and, West Africa known as the yam belt region of the world is regarded as centre of diversity of a key yam pathogen [28]. The severity of *fungi* infection in an area is usually based on a host cultivar cum pathogen strain interaction and predicated on a combination of factors including genetic (its rapid evolution) and favorable environmental conditions [27].

## Conclusion

In this study, seven isolates of *C. alatae*, Ca5, Ca14, Ca16, Ca22, Ca24, Ca32 and Ca34, and one isolate of *L. theobromae*, Lt1 were identified to cause anthracnose disease of yam in the Cross River yam farming areas. To the best of our knowledge, this is the first time that *L. theobromae* has been isolated from diseased yam leaves in Nigeria. The outcome of this study is a pointer to a mix infection of different fungal pathogens and the enormity of field losses that may have been incurred by farmers on a yearly basis over time and the impact on food security occasioned by anthracnose disease in this zone. These findings could serve as a lunch pad for plant breeders and molecular biologists, specifically, genomic specialists to develop strategies to produce *D. alata* and *D. rotundata* resistant to anthracnose in Nigeria and elsewhere. A success in this direction will boost the confidence of yam farmers to venture more into yam farming even at mechanized levels.

## Supporting information

**S1 Table. Primers sets used for identification of the fungal isolates.**
(DOCX)

**S2 Table. Percentage nucleotide similarity among the fungal isolates from yam.**
(DOCX)

**S1 Raw images.**
(PDF)

## Acknowledgments

The authors wish to thank Mfon Okon Akpan and Julius Oyohosuho Phillip for their technical support.

## Author Contributions

**Conceptualization:** Edak Aniedi Uyoh, Ene-Obong Effiom Ene-Obong.

**Data curation:** Nkese Ime Okon, Aniedi-Abasi Akpan Markson, Ekeng Ita Okon, Effiom Eyo Ita, Valentine Otang Ntui.

**Formal analysis:** Valentine Otang Ntui.

**Funding acquisition:** Ene-Obong Effiom Ene-Obong, Valentine Otang Ntui.

**Investigation:** Nkese Ime Okon, Aniedi-Abasi Akpan Markson, Ekeng Ita Okon, Effiom Eyo Ita.

**Methodology:** Nkese Ime Okon, Aniedi-Abasi Akpan Markson, Valentine Otang Ntui.

**Project administration:** Edak Aniedi Uyoh.

**Supervision:** Edak Aniedi Uyoh, Ene-Obong Effiom Ene-Obong, Valentine Otang Ntui.

**Writing – original draft:** Edak Aniedi Uyoh, Valentine Otang Ntui.

**Writing – review & editing:** Edak Aniedi Uyoh, Valentine Otang Ntui.

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
