## [Decision Letter · Decision Letter 0]

24 Jan 2022

PONE-D-21-40096Characterization of Colletotrichum alatae causing yam anthracnose in Cross River State, NigeriaPLOS ONE

Dear Dr. Otang

Thank you for submitting your manuscript to PLOS ONE. After careful consideration, we feel that it has merit but does not fully meet PLOS ONE’s publication criteria as it currently stands. Therefore, we invite you to submit a revised version of the manuscript that addresses the points raised during the review process.

ACADEMIC EDITOR: Reviewers raised several valid questions in the manuscript and provided useful comments. Therefore, I suggest to the authors follow the reviewer's comments to modify the manuscript. Recommended for major revision.

We look forward to receiving your revised manuscript.

Kind regards,

Karthikeyan Adhimoolam

Academic Editor

PLOS ONE

Journal Requirements:

[This research was supported by the Nigerian Tertiary Education Trust Fund (TETFund) grant TETFUND/DR&D/CE/NRF/2016/STI/13/VOL.1 awarded to the Department of Genetics and Biotechnology, University of Calabar, Calabar, Nigeria, for "Production of anthracnose-resistant yam seedlings for use by Nigerian farmers.]

 [This research was supported by the Nigerian Tertiary Education Trust Fund (TETFund) grant TETFUND/DR&D/CE/NRF/2016/STI/13/VOL.1 awarded to the Department of Genetics and Biotechnology, University of Calabar, Calabar, Nigeria, for "Production of anthracnose-resistant yam seedlings for use by Nigerian farmers. ]

5. We note that Figure 1 in your submission contain map images which may be copyrighted. All PLOS content is published under the Creative Commons Attribution License (CC BY 4.0), which means that the manuscript, images, and Supporting Information files will be freely available online, and any third party is permitted to access, download, copy, distribute, and use these materials in any way, even commercially, with proper attribution. For these reasons, we cannot publish previously copyrighted maps or satellite images created using proprietary data, such as Google software (Google Maps, Street View, and Earth). For more information, see our copyright guidelines: http://journals.plos.org/plosone/s/licenses-and-copyright.

a) You may seek permission from the original copyright holder of Figure 1 to publish the content specifically under the CC BY 4.0 license.  

Additional Editor Comments:

Reviewers raised several valid questions in the manuscript and provided useful comments. Therefore, I suggest to the authors follow the reviewer's comments to modify the manuscript. Recommended for major revision.

Reviewers' comments:

Reviewer's Responses to Questions

**Comments to the Author**

1. Is the manuscript technically sound, and do the data support the conclusions?

Reviewer #1: Yes

Reviewer #2: Partly

2. Has the statistical analysis been performed appropriately and rigorously? 

Reviewer #1: Yes

Reviewer #2: No

3. Have the authors made all data underlying the findings in their manuscript fully available?

Reviewer #1: Yes

Reviewer #2: Yes

4. Is the manuscript presented in an intelligible fashion and written in standard English?

Reviewer #1: Yes

Reviewer #2: No

5. Review Comments to the Author

Reviewer #1: Dear Author

Your submission entitled "Characterization of Colletotrichum alatae causing yam anthracnose in Cross River State, Nigeria" has been reviewed.

Kindly resubmit the manuscript with corrections in track changes mode, the other one, with clear manuscript).

Regards,

Reviewer #2: Comments

The manuscript entitled “Characterization of Colletotrichum alatae causing yam anthracnose in Cross River State, Nigeria” deals with the characterization of the Yam anthracnose pathogen through morphological, cultural, and molecular methods. Eight isolates of Colletotrichum spp. was obtained and sequence analysis revealed that the isolates are distinct. Screening the landraces with the virulent isolate, showed differential reaction which will help in breeding yam for anthracnose resistance. The paper provides information about the Yam pathogen in Cross river State, Nigeria, which is important to understand the etiology of the pathogen and the spread of the disease within the community. The diversity of C. alatae is said to result from its high potential for gene flow; and its virulence, might be due to recombination of its virulence alleles. However, some data should be appended and re-oriented to improve the quality of the paper. Overall, the English language can be improved and there are some grammatical errors which need to be corrected.

Abstract:

The abstract is written well. However, some of the data are misleading in screening the landraces which are repeated esp . … 15.15% were moderately resistant (MS), which is wrongly represented. 18.18% were resistant while 18.18% were highly resistant (HR). The data is not clear.

Introduction:

Can be shortened and the importance of Yam can be reduced. Instead the authors can focus in the disease and the present control measures followed. Besides, the international status of the disease and pathogen can also be represented as described by other scientists.

Materials and Methods:

Should be clear and concise and the methodology should be described. The cultural and morphological characters used for identification were not mentioned in this section. The type and methods of survey carried out can be elaborated. While using a scale, you can follow percent disease index rather than incidence. Any previous reference is there for disease scale. Why data were recorded 16 weeks post inoculation? Whether the symptoms appeared only by that time? How long the plants were maintained in the glass house. In soil or in pot culture?. The concentration (spores/ml) of the inoculam should be mentioned. Statistical analysis followed should be included. How many replications were made for the screening experiment. How many plants were used for landraces/accessions for screening?

Results

The result section can be reduced and at present it is elaborate. Some queries which are observed in the result are

i) The figures and tables can be reduced. Some tables can be deleted viz., Table 5 and Table 6. Table 3 can be given as a supplementary file. Similarly, Table 7 & 8 can be clubbed and the disease incidence and reaction can alone be presented as a single table.

ii) Whether rapid growth in plate is directly proportional to the virulence can be explained.

iii) Whether any specific difference in mycelial growth pattern was observed. Whether Acervuli was observed in the isolate in the diseased specimens. You can document the data to see the difference. Any sexual fruiting body was observed in the cultures viz., Perithecia. Since it deals with differentiating Colletotrichum spp. these characters are important. No conidial photos of the isolates were presented. Is there any variation of the hyaline, single celled conidia of C. alatae with respect to C. gloeosporioides observed as there exist variation among the species of Colletotrichum.

iv) Amplification of ITS region cannot differentiate within species level. If there are any SCAR markers specific for C. gloeosporioides, it can be used to differentiate C. alatae.

v) Some RAPD and ISSR primers can be used to differentiate the isolates which can be used to identify the marker unique in C. alatae .

vi) The accession number to the sequenced isolate can be given instead of the sequence.

vii) The variation in the sequence with respect to C. gloeosporioides can be highlighted if the authors justify the portion of the genome which is variable.

viii) The sequence of CA11 presented in Table 5 if blasted in NCBI database does not any match to C. alatae. Rather it shows identity to Lasiodiplodia theobromae. Justify.

ix). Whether you have submitted the sequence in the NCBI or any other database and the accession number obtained is not clear.

x). Disease incidence of 41-60 % is given as moderately resistance which looks very high. If the plant is 60 % infected how can we place in the resistant category. Pl. refer the reference for the reaction scale.

Discussion

It can be reduced to a certain extent and the main point alone can be discussed. However, the authors have substantially discussed the variation within the pathogen that may have arised due to recombination/gene flow.

Conclusion :

Conclusion should directly depict the results from the manuscript and its application in future can be highlighted.

6. PLOS authors have the option to publish the peer review history of their article (what does this mean?). If published, this will include your full peer review and any attached files.

Reviewer #1: **Yes: **Nasir Ahmed Rajput Assistant Professor, Department of Plant Pathology, University of Agriculture, Faisalabad Pakistan

Reviewer #2: No

---

## [Author Response · Author response to Decision Letter 0]

26 Feb 2022

Comment: Please ensure that your manuscript meets PLOS ONE's style requirements, including those for file naming. The PLOS ONE style templates can be found at 

Response: Manuscript formatted accordingly

Comment: We note that you have provided funding information that is currently declared in your Funding Statement. However, funding information should not appear in the Acknowledgments section or other areas of your manuscript. We will only publish funding information present in the Funding Statement section of the online submission form. 

 [This research was supported by the Nigerian Tertiary Education Trust Fund (TETFund) grant TETFUND/DR&D/CE/NRF/2016/STI/13/VOL.1 awarded to the Department of Genetics and Biotechnology, University of Calabar, Calabar, Nigeria, for "Production of anthracnose-resistant yam seedlings for use by Nigerian farmers.

Response: We have removed the funding statement from the manuscript. Please publish the funding statement as it is.

Comment: PLOS ONE now requires that authors provide the original uncropped and unadjusted images underlying all blot or gel results reported in a submission’s figures or Supporting Information files.

Response: We have included the original uncropped and unadjusted gel images.

Comment: Please include your full ethics statement in the ‘Methods’ section of your manuscript file. In your statement, please include the full name of the IRB or ethics committee who approved or waived your study, as well as whether or not you obtained informed written or verbal consent. If consent was waived for your study, please include this information in your statement as well. 

Response: We have included ethical statement in materials and methods

Comment: We note that Figure 1 in your submission contain map images which may be copyrighted.

Response: Figure 1 has been deleted.

To Reviewer 1

Comment: What were the key findings from the characterization of the pathogen?

Response: Eight isolates, Ca5, Ca11, Ca14, Ca16, Ca22, Ca24, Ca32 and Ca34 of C. alatae were found to be associated with yam infection in Cross River State, with Ca11 as the most prevalent, occurring in all the locations. These isolates were classified into four forms which included the slow-growing grey (SGG), the fast-growing grey (FGG), the fast-growing salmon (FGS), and the fast-growing olive (FGO).

Comments: This section seems to be verbose and should be discussed briefly.

Response: We have expunged some sentences

Comment: Is the pathogen infection issue in all form of Yam cultivars? And what about disease percent?

Response: This has been described in the introduction as follows: In Nigeria, Colletotrichum disease complex (commonly referred to as anthracnose or die-back) remains one of the most challenging and destructive diseases, causing heavy losses in yam. Anthracnose has been implicated in yam tuber yield losses ranging from 50 to 90% under favorable conditions for pathogen infection, establishment, and disease development

Comment: Objectives of this study not clearly mentioned

Response: We have rephrased the objective to: The objective of this study was to characterize and identify Colletotrichum isolates associated with yam anthracnose in Cross River State, Nigeria and to determine the relationship among them as well as their virulence.

Comment: The author must add the reference of isolation protocol

Response: The reference [16] has been added

Comment: Was the molecular characterization of any help in finding out any certain gene which can be used for further evaluating the infection capabilities of this pathogen?

Response: The molecular characterization was based on the ITS region. We did not do a whole genome sequence which would have revealed other genes which could be helpful in evaluating the infection capabilities of the isolates.

Comments: This section should be more clear and suggested for revision.

Response: We revised the DNA extraction protocol accordingly

Comment: Too much typo error. Use small case for chemicals name 

Response: We have corrected all the typo errors.

Comment: During the characterization phase the phylogenetic analysis gave any other major infecting fungal species in closely clustered clade and what does it says overall for future of fungal infestation in Nigeria’s agriculture fields?

Response: Yes, there were some other Colletotrichum species found in same clade with some of our isolates, but quite distinct in terms of nucleotide identity. This however, indicates there could be a mixed infection of yam in the field. We have added this phrase in the discussion.

Comment: Did the author repeat this experiment for the confirmation of results?

Response: We did not repeat the experiment. This was because we also did whole plant bioassay. Besides, the pathogenicity test was done in very sterile condition without any contamination

Comment: Correct equation format suggested 

Response: This has been corrected as suggested

Comment: Give specific name to the primer set rather than 1 or 2

Response: This has been corrected

Comment: This section (discussion) is too verbose which makes difficulty for the readers. The author suggested to concise it with necessary information and literature list.

Response: This has been done as suggested 

Comment: What are ultimate domestic market and export market benefits of your studies?

Response: This study will be useful to yam breeders in Nigeria and elsewhere. Please see our conclusion.

To Reviewer 2

 Abstract

Comment: The abstract is written well. However, some of the data are misleading in screening the landraces which are repeated esp . … 15.15% were moderately resistant (MS), which is wrongly represented. 18.18% were resistant while 18.18% were highly resistant (HR). The data is not clear.

Response: We have corrected the sentence to: Inoculation of 20 D. alata and 13 D. rotundata landraces with isolate Ca11, showed that 63.64% of the landraces were susceptible while 36.36% were resistant

Introduction:

Comment: Can be shortened and the importance of Yam can be reduced. Instead the authors can focus in the disease and the present control measures followed. Besides, the international status of the disease and pathogen can also be represented as described by other scientists.

Response: This has been done as suggested

Materials and Methods:

Comment: Should be clear and concise and the methodology should be described. The cultural and morphological characters used for identification were not mentioned in this section. The type and methods of survey carried out can be elaborated. While using a scale, you can follow percent disease index rather than incidence. Any previous reference is there for disease scale. Why data were recorded 16 weeks post inoculation? Whether the symptoms appeared only by that time? How long the plants were maintained in the glass house. In soil or in pot culture?. The concentration (spores/ml) of the inoculum should be mentioned. Statistical analysis followed should be included. How many replications were made for the screening experiment. How many plants were used for landraces/accessions for screening?

Response: Please see the responses below, these have also been incorporated in the manuscript

Cultural and morphological characteristics by which the isolates were identified include mycelia colour, growth pattern, nature of mycelia and speed of growth in the Petri dish. All these characters were done through visual appreciation.

i) The sampling method used was cluster sampling. The local government areas were the clusters where farms were identified and sampled in selected villages (units) within the Local Government areas. Simple random sampling procedures were used in selecting the villages in each of the Local Government Areas. As such each of the villages had equal chances of being selected since all were known to be major yam-cultivating area. Hence six Local Government Areas (six clusters) were used and villages with history of yam cultivation were selected randomly to arrive at the villages sampled per Local Government Area.

ii) Symptoms started appearing at 8 weeks post inoculation and progressed slowly, so we allowed the symptoms to develop fully before taking data at 16 weeks.

iii) Plants were maintained for 6 months in bags. 

iv) Spore concentration used was 1.0 x 104 spore/ml throughout the study.

v) During field screening, three replicates were maintained.

vi) 27 Landraces and 6 accessions were screened. Each was replicated thrice (3 plants per landrace/accession).

Results

The result section can be reduced and at present it is elaborate. Some queries which are observed in the result are

Comment: The figures and tables can be reduced. Some tables can be deleted viz., Table 5 and Table 6. Table 3 can be given as a supplementary file. Similarly, Table 7 & 8 can be clubbed and the disease incidence and reaction can alone be presented as a single table.

Response: The figures have been reduced to 5. The number of Tables have been reduced to 4. We have deleted Table 5. We have taken Tables 3 and 6 to supplementary file as supplementary Tables 1 and 2. We combined Tables 7 and 8 as one Table which is now Table 4.

Comment: Whether rapid growth in plate is directly proportional to the virulence can be explained.

Response: The rapid growth on the plates does not directly transcribe to virulence as the isolate Ca 11 which is one of the slowest growing (Table 3) is the most virulent among the isolates studied.

Comment: Whether any specific difference in mycelial growth pattern was observed. 

Response: Differences in mycelia growth pattern were also observed. This ranged from small to large concentric rings exhibiting mostly cottony growth mycelia (Table 3).

Comment: Whether Acervuli was observed in the isolate in the diseased specimens. You can document the data to see the difference. 

Response: Other characteristics such as Acervuli were observed in the plates but not in infected specimens. These had dark spines (setae) at the edge of the structure and among the conidiophores (Fig.3j)

Comment: Any sexual fruiting body was observed in the cultures viz., Perithecia. Since it deals with differentiating Colletotrichum spp. these characters are important. 

Response: Perithecia were not observed

Comment: No conidial photos of the isolates were presented. 

Response: Photos of the two types of conidia encountered have been added (Fig. 3i)

Comment: Is there any variation of the hyaline, single celled conidia of C. alatae with respect to C. gloeosporioides observed as there exist variation among the species of Colletotrichum.

Response: there were no distinct differences in the conidial appearance in terms of size and shape. All were single -celled, oblong or cylindrical, broadly rounded at both ends and some slightly tapering to the base (Fig. 3i)

Comment: Amplification of ITS region cannot differentiate within species level. If there are any SCAR markers specific for C. gloeosporioides, it can be used to differentiate C. alatae.

Response: We also intended to use some molecular markers to confirm the identification of the isolates. However, the project ended and we did not have funds to continue with the work. We are however sourcing for funds, as soon as we get some funding we will do detail molecular analysis of the isolates, using some molecular markers.

Comment: Some RAPD and ISSR primers can be used to differentiate the isolates which can be used to identify the marker unique in C. alatae.

Response: As explained already, we intended to use some molecular markers to confirm the identification of the isolates. However, the project ended and we did not have funds to continue with the work. We are however sourcing for funds for another level of study on these isolates using these primers.

Comment: The accession number to the sequenced isolate can be given instead of the sequence.

Response: This has been done

Comment: The variation in the sequence with respect to C. gloeosporioides can be highlighted if the authors justify the portion of the genome which is variable.

Response: We have highlighted this in the text as follows: The 5.8s rRNA was the most conserved genomic region across the isolates and the C. gloeosporioides reference genome with minimal variations, whereas the ITS1 and ITS2 were the most diverse.

Comment: The sequence of CA11 presented in Table 5 if blasted in NCBI database does not any match to C. alatae. Rather it shows identity to Lasiodiplodia theobromae. Justify.

Response: We have justified this in the discussion as follows: Isolate Lt1 was the most virulent, but it was one of the slow growing types in the plates, indicating that rapid growth in the plate is not proportional to virulence. A blast of this isolate on NCBI shows <75% nucleotide identity with other C. gloeosporioides, and >97% nucleotide identity with many Lasiodiplodia theobromae isolates. L. theobromae is reported to be cosmopolitan in nature and has been reported to cause dieback infections on cash crops such as cocoa and yams [54]. This and the fact that the isolate clustered in the same clade with L. theobromae supported by high boostrap value (Fig. 5), confirmed its identity. Therefore, we named it L. theobromae isolate Lt1 with the accession number OM365423. When yam landraces/accessions were challenged with this isolate, a high number of landraces were susceptible to the pathogen suggesting L. theobromae infects yam. More studies are required to evaluate the prevalence of L. theobromae infection of yam in Nigeria.

Comment: Whether you have submitted the sequence in the NCBI or any other database and the accession number obtained is not clear.

Response: We have included a statement on this in the results as follows: The nucleotide sequences of the ITS region for six of the isolates were submitted to National Centre for Biotechnology Information (NCBI) database and were assigned the following accession numbers: OM365422 (Ca5), OM365423 (Lt1), OM365424 (Ca14), OM365425 (Ca24), OM365426 (Ca32) and OM427500 (Ca34). 

Comment: Disease incidence of 41-60 % is given as moderately resistance which looks very high. If the plant is 60 % infected how can we place in the resistant category. Pl. refer the reference for the reaction scale.

Response: We have corrected this to show that disease indices above 40% are considered to be susceptible (Table 4)

Discussion

Comment: It can be reduced to a certain extent and the main point alone can be discussed. However, the authors have substantially discussed the variation within the pathogen that may have arised due to recombination/gene flow.

Response: The discussion has been considerably reduced

Conclusion :

Comment: Conclusion should directly depict the results from the manuscript and its application in future can be highlighted.

Response: This has been corrected as suggested.

---

## [Decision Letter · Decision Letter 1]

22 Mar 2022

PONE-D-21-40096R1Characterization of Colletotrichum alatae causing yam anthracnose in Cross River State, NigeriaPLOS ONE

Dear Dr. Ntui,

Thank you for submitting your manuscript to PLOS ONE. After careful consideration, we feel that it has merit but does not fully meet PLOS ONE’s publication criteria as it currently stands. Therefore, we invite you to submit a revised version of the manuscript that addresses the points raised during the review process.

ACADEMIC EDITOR: Reviewers raised several valid questions in the manuscript and provided useful comments. Therefore, I suggest to the authors follow the reviewer's comments to modify the manuscript. Recommended for minor revision.

We look forward to receiving your revised manuscript.

Kind regards,

Karthikeyan Adhimoolam

Academic Editor

PLOS ONE

Journal Requirements:

Reviewers' comments:

Reviewer's Responses to Questions

**Comments to the Author**

1. If the authors have adequately addressed your comments raised in a previous round of review and you feel that this manuscript is now acceptable for publication, you may indicate that here to bypass the “Comments to the Author” section, enter your conflict of interest statement in the “Confidential to Editor” section, and submit your "Accept" recommendation.

Reviewer #1: All comments have been addressed

Reviewer #2: All comments have been addressed

2. Is the manuscript technically sound, and do the data support the conclusions?

Reviewer #1: Yes

Reviewer #2: Partly

3. Has the statistical analysis been performed appropriately and rigorously? 

Reviewer #1: Yes

Reviewer #2: N/A

4. Have the authors made all data underlying the findings in their manuscript fully available?

Reviewer #1: Yes

Reviewer #2: No

5. Is the manuscript presented in an intelligible fashion and written in standard English?

Reviewer #1: Yes

Reviewer #2: Yes

6. Review Comments to the Author

Reviewer #1: (No Response)

Reviewer #2: The authors have carried out all the corrections as suggested by the reviewer. Now the quality of the paper is improved. But there are many basic flaws which need to be addressed besides some minor corrections.

Minor corrections

Abstract : Spelling mistakes . Need to be carefully addressed throughout the manuscript. Lsidioplodia theobromae, (Lasiodiplodia theobromae).

In the third line include…In the present study, the pathogen…….

End of Introduction Proffer ….

The identity of the isolates in sequencing seems to be very less and should be clarified. Bec. < 75 % identity could not exist within a species. In that case, it should be a different genus.

None of the C. alatae isolates was in the same cluster with the reference sequence indicating that the isolates are distinct……Is not clear, If so how you claim it is C. alatae if it is distinct….Can be rewritten or reframed.

Major corrections

In the materials and methods in whole plant bioassay it is denotes as

Screening of yam for tolerance to C. alatae: Whole plant bioassay

In the above title, you have mentioned tolerance to C. alatae. But challenged with Lasiosiplodia theobromae ……The yam plants were challenged with isolate Lt1 which was the most virulent according to the pathogenicity test. ……….Then the entire experiment goes wrong as it is with a different pathogen. Even though it is highly pathogenic, it is a different pathogen with different spore characters. In that case, the paper should address the importance of two pathogens and not only C. alatae. Bec. The title deals with C. alatae. But the landraces/accessions are screened for L. bromae. The authors will be confused with these statements. Substantiate.

If Lt1 is used for screening the isolates how will you claim it is resistant to Colletotrichum spp? In that case, even the scale used should have a reference as it is general or specific for Colletotrichum pathogens.

In the table 4 in the legends it is still indicated that 41- 60 % is moderately resistant but in response to reviewers it is claimed that > 40 % is considered as susceptible. Clarify.

7. PLOS authors have the option to publish the peer review history of their article (what does this mean?). If published, this will include your full peer review and any attached files.

Reviewer #1: **Yes: **Dr. Nasir Ahmed Rajput, Department of Plant Pathology, University of Agriculture, Faisalabad Pakistan

Reviewer #2: No

---

## [Author Response · Author response to Decision Letter 1]

27 Apr 2022

Response to comments

Reviewer #2: The authors have carried out all the corrections as suggested by the reviewer. Now the quality of the paper is improved. But there are many basic flaws which need to be addressed besides some minor corrections.

Minor corrections

Comment: Abstract: Spelling mistakes . Need to be carefully addressed throughout the manuscript. Lsidioplodia theobromae, (Lasiodiplodia theobromae).

Response: This has been corrected in the abstract and throughout the manuscript

Comment: In the third line include…In the present study, the pathogen…….

Response: The phrase has been added.

End of Introduction Proffer ….

The identity of the isolates in sequencing seems to be very less and should be clarified. Bec. < 75 % identity could not exist within a species. In that case, it should be a different genus.

Response: Although the isolates had < 75% identity, they were identified as C. alata based on their cultural and morphological characteristics, mycelial growth rate as well as the fact that they clustered with other Colletotrichum species. 

Comment: None of the C. alatae isolates was in the same cluster with the reference sequence indicating that the isolates are distinct……Is not clear, If so how you claim it is C. alatae if it is distinct….Can be rewritten or reframed.

Response: The sentence has been rephrased to: None of the C. alatae isolates was in the same cluster with the reference sequence KC010547.1 but were found clustering with other C. gloeosporioides. 

Note: The reference sequence was used to design primers. It could be that the primers did bind our isolates, but the nucleotide identity is not the same. That is why the isolates are clustering with other C. gloeosporioides and not with the reference sequence.

Major corrections

Comment: In the materials and methods in whole plant bioassay it is denotes as

Screening of yam for tolerance to C. alatae: Whole plant bioassay

Response: We have corrected this to: Screening of yam for tolerance to Lasiodiplodia theobromae: Whole plant bioassay

Comment: In the above title, you have mentioned tolerance to C. alatae. But challenged with Lasiosiplodia theobromae ……The yam plants were challenged with isolate Lt1 which was the most virulent according to the pathogenicity test. ……….Then the entire experiment goes wrong as it is with a different pathogen.

Response: We have modified the title to: Characterization of some fungal pathogens causing anthracnose disease on yam in Cross River State, Nigeria

Comments: Even though it is highly pathogenic, it is a different pathogen with different spore characters. In that case, the paper should address the importance of two pathogens and not only C. alatae. Bec. The title deals with C. alatae. But the landraces/accessions are screened for L. bromae. The authors will be confused with these statements. Substantiate.

Response: We have modified the title and addressed Lasiodiplodia theobromae in the manuscript

If Lt1 is used for screening the isolates how will you claim it is resistant to Colletotrichum spp? In that case, even the scale used should have a reference as it is general or specific for Colletotrichum pathogens.

Response: We have modified the section to: Screening D. alata and D. rotundata for resistance to Lasiodiplodia theobromae in the screen house. The scale used is general for classification of disease resistance

Comment: In the table 4 in the legends it is still indicated that 41- 60 % is moderately resistant but in response to reviewers it is claimed that > 40 % is considered as susceptible. Clarify.

Response: We have corrected the legend as follows: Accessions or land races with a disease incidence of 0-20% were given a score of 1and considered as highly resistant (HR), those with a disease incidence of 21-40% were given a score of 2 and regarded as resistant (R), those with a disease incidence of 41-60% (3), and regarded as moderately susceptible (MS), those with a disease incidence of 61-80% (4), and regarded as susceptible (S), and those with a disease incidence of 81-100% (5), and regarded as highly susceptible (HS). In general, indices >40% are susceptible.

---

## [Decision Letter · Decision Letter 2]

14 Jun 2022

Characterization of some fungal pathogens causing anthracnose disease on yam in Cross River State, Nigeria

PONE-D-21-40096R2

Dear Dr. Valentine,

We’re pleased to inform you that your manuscript has been judged scientifically suitable for publication and will be formally accepted for publication once it meets all outstanding technical requirements.

Kind regards,

Karthikeyan Adhimoolam

Academic Editor

PLOS ONE

Additional Editor Comments (optional):

Reviewers' comments:

Reviewer's Responses to Questions

**Comments to the Author**

1. If the authors have adequately addressed your comments raised in a previous round of review and you feel that this manuscript is now acceptable for publication, you may indicate that here to bypass the “Comments to the Author” section, enter your conflict of interest statement in the “Confidential to Editor” section, and submit your "Accept" recommendation.

Reviewer #3: (No Response)

2. Is the manuscript technically sound, and do the data support the conclusions?

Reviewer #3: Yes

3. Has the statistical analysis been performed appropriately and rigorously? 

Reviewer #3: N/A

4. Have the authors made all data underlying the findings in their manuscript fully available?

Reviewer #3: Yes

5. Is the manuscript presented in an intelligible fashion and written in standard English?

Reviewer #3: No

6. Review Comments to the Author

Reviewer #3: (No Response)

7. PLOS authors have the option to publish the peer review history of their article (what does this mean?). If published, this will include your full peer review and any attached files.

Reviewer #3: No

---

## [Editor Report · Acceptance letter]

20 Jun 2022

PONE-D-21-40096R2 

Characterization of some fungal pathogens causing anthracnose disease on yam in Cross River State, Nigeria 

Dear Dr. Ntui:

I'm pleased to inform you that your manuscript has been deemed suitable for publication in PLOS ONE. Congratulations! Your manuscript is now with our production department. 

Kind regards, 

on behalf of

Dr. Karthikeyan Adhimoolam 

Academic Editor

PLOS ONE